# Flavones: The Apoptosis in Prostate Cancer of Three Flavones Selected as Therapeutic Candidate Models

**DOI:** 10.3390/ijms24119240

**Published:** 2023-05-25

**Authors:** Se Hyo Jeong, Hun Hwan Kim, Min Young Park, Pritam Bhagwan Bhosale, Abuyaseer Abusaliya, Chung Kil Won, Kwang Il Park, Eunhye Kim, Jeong Doo Heo, Hyun Wook Kim, Meejung Ahn, Je Kyung Seong, Gon Sup Kim

**Affiliations:** 1Department of Veterinary Medicine, Research Institute of Life Science, Gyeongsang National University, 501 Jinju-daero, Jinju 52828, Republic of Korea; tpgy123@gmail.com (S.H.J.); shark159753@naver.com (H.H.K.); lilie17@daum.net (M.Y.P.); shelake.pritam@gmail.com (P.B.B.); yaseerbiotech21@gmail.com (A.A.); wonck@gnu.ac.kr (C.K.W.); kipark@gnu.ac.kr (K.I.P.); eunhyekim@gnu.ac.kr (E.K.); 2Biological Resources Research Group, Gyeongnam Department of Environment Toxicology and Chemistry, Korea Institute of Toxicology, 17 Jegok-gil, Jinju 52834, Republic of Korea; jdher@kitox.re.kr; 3Division of Animal Bioscience & Integrated Biotechnology, Jinju 52725, Republic of Korea; hwkim@gnu.ac.kr; 4Department of Animal Science, College of Life Science, Sangji University, Wonju 26339, Republic of Korea; meeahn@sangji.ac.kr; 5Laboratory of Developmental Biology and Genomics, BK21 PLUS Program for Creative Veterinary Science Research, Research Institute for Veterinary Science, College of Veterinary Medicine, Seoul National University, Seoul 08826, Republic of Korea; snumouse@snu.ac.kr

**Keywords:** flavonoids, flavones, prostate cancer, cancer cell death, apoptosis

## Abstract

Cancer is a widespread but dangerous disease that can strike anyone and is the second 1leading cause of death worldwide. Prostate cancer, in particular, is a prevalent cancer that occurs in men, and much research is being done on its treatment. Although chemical drugs are effective, they have various side effects, and accordingly, anticancer drugs using natural products are emerging. To date, many natural candidates have been discovered, and new drugs are being developed as drugs to treat prostate cancer. Representative candidate compounds that have been studied to be effective in prostate cancer include apigenin, acacetin and tangeretin of the flavone family among flavonoids. In this review, we look at the effects of these three flavones on prostate cancer cells via apoptosis in vitro and in vivo. Furthermore, in addition to the existing drugs, we suggest the three flavones and their effectiveness as natural anticancer agents, a treatment model for prostate cancer.

## 1. Introduction

### 1.1. Cancer Causes and Features

Cancer is one of the world’s worst forms of disease, which is expensive and complex to treat and has side effects from chemicals. There are many different cancer causes, which can be divided into intrinsic and extrinsic risk factors. Accidental errors in DNA replication cause intrinsic risk, which cannot be fixed. Extrinsic risk factors include endogenous and exogenous risk factors. Endogenous risk factors that can be partially modified include aging, hormones, growth factors, inflammation and DNA repair mechanisms. Exogenous risk factors can be modulated, including radiation, chemical carcinogens, oncogenic viruses and smoking [1]. Characteristics of cancer cells include sustained maintenance of proliferative signals, evasion of growth inhibitory agents, induction of angiogenesis, activation of invasion and metastasis, and the potential for permanent proliferation. Abnormalities in cellular energy metabolism, immune disruption avoidance, tumor-promoting inflammation, genomic instability and mutations have also been suggested. Additionally, recently, phenotypic plasticity unlocking, non-mutant epigenetic reprogramming, polymorphic microbiome and senescent cells are emerging as a hallmark of cancer [2]. Especially apoptosis is a necessary process that plays a role in regulating cancer. A malfunction of apoptosis is also a reason for cancer, so the cancer cell with an apoptotic problem does not follow this mechanism [3].

### 1.2. The Relationship between Causes of Prostate Cancer and Androgen Receptors

Among numerous cancers, prostate cancer is one of the most common cancers in men, with the number of patients increasing rapidly due to the aging of the population and the influence of diet [4,5,6]. The average development and maintenance of the prostate depend on androgens acting through androgen receptors (AR). Therefore, androgens and androgen receptors play an essential regulatory role in prostate cancer progression [7]. For this reason, the androgen receptor is a significant therapeutic target for prostate cancer [8]. When a man is diagnosed with prostate cancer, treatment options include surgical prostate removal, chemotherapy and radiation therapy. Although these treatments are effective in the early stages of cancer, they are not very effective in prostate cancer, which has progressed significantly [9]. Rather, prostate cancer metastasizes more easily when testosterone is blocked with commonly known androgen-suppressing hormone therapy [10]. Due to these problems, many studies are being conducted on anticancer therapies using flavonoids and natural bioactive compounds [11,12]. Against prostate cancer, cancer cell death through apoptosis, especially anticancer action using natural bioactive compounds, such as flavonoids, has played an important role [13].

### 1.3. Flavonoids

The most prevalent chemicals found in plants and fungi, including vegetables and fruits, are flavonoids, secondary metabolites plants produce for self-protection from external stimuli. Six subclasses of flavonoids can be distinguished: flavones, flavan-3-ols, flavanones, flavonols, anthocyanidins and isoflavones [14,15]. Among them, more than 6000 flavonoids have been found [16]. According to many studies, these flavonoids are expected to have a significant positive impact on cancer therapy [17].

### 1.4. Functions of Flavones and Their Effect on Prostate Cancer

One of the subclasses of flavonoids, flavones are found in various plants, including red pepper, parsley, celery and many other plants. Flavone has the chemical formula C6–C3–C6, and the molecular formula is C_15_H_10_O_2_. It has a double bond between C2 and C3 from the basic flavonoid structure and a ketone group at C4 [18,19] (Figure 1A). Researchers discovered that flavones are involved in a wide range of biological processes, including protecting cell membranes [20,21], acting as antioxidants [22,23,24], antivirals [25], antibacterials [26] and treating prostate hyperplasia [27,28]. In particular, apigenin, acacetin and tangeretin are three flavones beneficial for prostate cancer [29,30,31,32] (Figure 1B–D).

## 2. Prostate Cancer

### 2.1. Prostate Cancer Epidemiology and Mortality in Humans

In humans, prostate cancer is the second most common cancer and the fifth leading cause of death worldwide. One of the reasons for the high mortality rate is that many cases are asymptomatic in the early stages, and unless special measures are taken, the course of cancer worsens [33]. In particular, elderly patients have a high mortality rate accompanied by the incidence rate, which is greatly influenced by diet, lifestyle, race, genetic causes and age [34,35].

For dietary reasons, a higher intake of saturated animal fats increases the level of androgens, which increases the risk of prostate cancer [36,37]. Excessive intake of calcium and milk has also been related to prostate cancer [38]. A deficiency of vitamin D increases the risk of prostate cancer [39,40]. Excessive alcohol consumption increased the incidence of prostate cancer, and coffee significantly lowered the risk of advanced prostate cancer [41,42,43,44]. Supplements, such as selenium, reduce the risk of prostate cancer, and coffee significantly lowers the risk of advanced prostate cancer [39,40,41,42,43,44].

It is known that about 20% of family history is due to prostate cancer. Among them, a gene called HPC1 encodes an enzyme involved in interferon-mediated anti-inflammatory transmission, which reduces antiviral activity and regulates apoptosis to control prostate cancer [45,46,47,48]. In men with BRCA1 and 2 mutations in aggressive prostate cancer, the role of HPC sub-genes plays an important role, and BRCA2 mutations are associated with an increased incidence of prostate cancer. PALB2 is a protein that interacts with BRCA2 and is closely related to familial prostate cancer [49,50].

In the multi-ethnic United States, American Indians and Alaskans (46.9), Asian and Pacific Islanders (52.4) and Caucasians (93.9) had lower rates of prostate cancer [33]. African Americans (157.6) have a mutation on chromosome 8q24 that other races do not have; this increases the risk of prostate cancer [51]. Also, African Americans are more likely to have mutations in tumor suppressor genes, such as EphB2, or apoptosis genes, such as Bcl-2 [52,53]. In particular, significantly higher PSA levels were found in black men than in white men [54].

### 2.2. Classification of Prostate Cancer

There are several types of prostate cancer. The most common is prostate adenocarcinoma, which occurs in the glands surrounding the prostate testicles and is present in all prostate cancer patients. The patients (between 2% and 4%) with transitional cell carcinoma of the prostate arise from cells connecting the urethra. In addition to this, there is a type that occurs in the squamous cells of the prostate, and there is also a small-cell prostate cancer characterized by growing faster than other types [55,56].

In the classification of prostate cancer, graded pathology is defined by the Gleason sum score on a morphological basis [57]. This score is the primary method for grading prostate cancer tissue and is an important prognostic factor. Accordingly, it is divided again according to the presence of bone metastases, resistance to androgen ablation therapy and chemotherapy [58]. Beyond the classification problem, according to these simple forms, treatment approaches are proposed and classified based on the microenvironment, such as androgen receptors, oncogenes and tumor suppressors [59].

The general criteria for high-risk prostate cancer set by the American Urological Association are a preoperative Gleason score of 8–10, PSA > 20 ng/mL and clinical stage ≥ T2c (TNM classification, T2: tumor size 2–5 cm). However, the need for a detailed classification system arose due to the inaccuracy of determining the T stage and the difference from the actual high risk according to the score. Therefore, clinical trials of new designs and classifications are needed to optimize the treatment of truly high-risk patients [60].

### 2.3. Prostate Cancer Causes and Treatment

Prostate cancer is a prevalent cancer in men. In normal prostate tissue, signaling pathways are regulated by androgens, maintaining a balance between prostate cell growth and apoptosis [8]. In problematic tissues, the balance is disrupted, and the expression of androgens leads to prostatic cell growth and pathological problems, resulting in prostate hyperplasia and, eventually, cancer [61]. “Androgen-dependent” treatment, which surgically blocks and controls these androgens, may be effective in early prostate cancer but eventually develops into “androgen-independent” cancer that is ineffective and continues to grow and metastasize [62,63,64]. The Androgen receptor (AR) plays a pivotal role in prostate cancer. Generally, androgens bind to AR and isolate chaperone proteins, including members of the heat shock protein family HSP27 and HSP70 [65]. It is sensitive to the increase in AR mRNA, even when androgens are blocked, and it is easy to bind with other male hormones [66,67,68]. In this case, the androgen-blocking method is useless, and the prostate-specific antigen (PSA) also comes out at a high level [69,70]. Unfortunately, it becomes castration-resistant prostate cancer (CRPC) [71].

Androgen receptor inhibition is significant in the treatment of prostate cancer. Anti-androgen drugs play a role here [72]. Previously, numerous first-generation anti-androgens were discovered, and second-generation anti-androgens existed. First-generation anti-androgens to inhibit AR used steroid analogs to block ligand activation [73]. However, it soon became resistant to various mutations and was no longer suitable. Second-generation anti-androgen drugs developed after that are more effective in treating CRPC, have increased specific binding to androgen receptors than steroid receptors, do not cause withdrawal symptoms and have significantly contributed to effective prostate cancer treatment [74].

Representative second-generation anti-androgen therapies include apalutamide (Figure 1E) and enzalutamide (Figure 1F). Androgens are converted to dihydrotestosterone (DHT) by 5-alpha reductase, and DHT binds to the androgen receptor, which is inhibited by both drugs [65] (Figure 2). Enzalutamide inhibits AR DNA binding, coactivator recruitment and AR nuclear translocation. Apalutamide, developed after, is a nonsteroidal anti-androgen that acts as an inhibitor of the ligand binding domain of the androgen receptor. This binding inhibits translocation to the nucleus and DNA binding. In short, it is an anti-androgen agent that acts as an androgen receptor antagonist, which is the main target of androgens, male hormones such as testosterone and dihydrotestosterone (DHT) [75]. It is more effective than enzalutamide, and blood-brain barrier permeability is reduced to decrease the risk of seizures. The FDA approved this drug in February 2018 [74,76].

### 2.4. Side Effects of the Mentioned Chemical Drugs (Apalutamide and Enzalutamide)

Apalutamide and enzalutamide still have side effects just as most chemicals, such as hair loss, anorexia, mouth sores, diarrhea or constipation, nausea and vomiting and fatigue [77,78]. In particular, enzalutamide has side effects, such as fatigue, constipation and seizures, that may occur [79,80]. Apalutamide also has common side effects, such as anemia, leukopenia, lymphopenia, high blood pressure, diarrhea, nausea, loss of appetite, fatigue, skin rash and less common side effects, such as ischemic heart disease, heart failure, pruritus and hypothyroidism. Also, it cannot be prescribed during pregnancy and is not recommended for seizure patients [78].

## 3. The Mechanism of Apoptosis in Prostate Cancer

Cancer, such as prostate cancer, undergoes anticancer control through various mechanisms. The most representative is apoptosis, which regulates cancer through mechanisms that arrest the cell cycle or prevent metastasis or proliferation [81].

### 3.1. Apoptosis Pathways

If apoptosis does not occur properly in cancer cells, the cells will proliferate indefinitely, invade and metastasize to other tissues and show a poor prognosis [81]. As such, apoptosis is considered necessary in cancer cells, and there are two pathways for apoptosis. One is the intrinsic pathway, and the other is the extrinsic pathway [82] (Figure 3).

The intrinsic pathway is mitochondria-mediated apoptosis, and cytochrome c release from mitochondria by the Bcl-2 family combines with ATP, apoptosis protease activation factor-1 (APAF-1) and pro-caspase-9 to form an apoptotic complex. Afterward, caspase-9, caspase-3, -6 and -7 are activated to cause apoptosis [82,83,84]. In addition, signaling pathways, such as PI3K/AKT, and various apoptosis regulatory proteins, such as P53, induce pro-apoptotic proteins (BAX, BAK, BIM, NOXA) and inhibit anti-apoptotic proteins (Bcl-2, Bcl-xL). These proteins are concerned with the intrinsic pathway [85,86].

The extrinsic pathway is an apoptosis pathway induced by death receptors, such as tumor necrosis factor receptor (TNF), TNF-related apoptosis-inducing ligand receptor (TRAIL) and Fas receptor [81]. Fas L or TNF ligands bind to death domains, such as TNF receptor-associated death domain (TRADD) and Fas-associated death domain (FADD), to form a death-inducing signaling complex (DISC), and caspase-8 and caspase-3 activate and cause apoptosis. In addition, these receptor-mediated pathways are also concerned with the intrinsic pathway to cause apoptosis [87,88,89].

### 3.2. Apoptosis in Prostate Cancer

As in other cancer, in prostate cancer, the interaction between cell growth, proliferation factors and apoptosis factors is considered necessary for the growth of the prostate. In prostate cancer cells, when these interactions, that is, problems with apoptosis regulation, are received, they lead to apoptosis avoidance, differentiation and infinite proliferation. To know the effective apoptosis mechanisms in prostate cancer progression requires studies integrating numerous clinical variables and biochemical markers [90,91].

Regarding the intrinsic and extrinsic apoptosis pathways in prostate cancer, the extrinsic pathway is preferentially triggered by death receptors and activates caspase-8, which directly activates caspase-3 to induce apoptosis. In the intrinsic pathway, caspase-9 is activated by cytochrome c release from mitochondria, which anti-apoptotic regulators of the Bcl family can inhibit. Lastly, caspase-3 is activated to cause apoptosis. In this mechanism, bioactive compounds, such as flavones, are important dietary regulators in causing this apoptosis [91].

### 3.3. The Genetic Mutations of Apoptosis in Prostate Cancer

A gene mutation causes fatal prostate cancer. It includes changes in amplification and biosynthesis, such as point mutations in LBD, the ligand-binding domain of AR, or overexpression of AR [64]. In addition, mutations in the RNase L gene, a hereditary prostate cancer 1 (HPC1) allele, inhibit the apoptosis pathway of prostate cancer [92].

The tumor suppressor gene p53, most commonly associated with apoptosis, regulates both apoptosis and the cell cycle in response to numerous cellular stresses, such as DNA or free radical damage. Through the upregulation of BAX in mitochondria, P53 has the ability to induce apoptosis, and this mutation of P53 inhibits apoptosis, helping cancer cells to proliferate indefinitely [93,94]. In prostate cancer, p53 mutations are uncommon in the early, well-differentiated state but become more frequent as metastatic disease or hormone-independent tumors progress [95].

## 4. Anticancer Effects of Three Selected Flavones in Prostate Cancer

### 4.1. Flavones as Promising Biochemical Agents for Anti-Prostate Cancer

Anticancer effects using flavones have been revealed through various studies and, especially, there are few evidences that the flavones are effective in prostate cancer. Among many flavones, apigenin, acacetin and tangeretin are known to be effective against prostate cancer [15,29,31,32].

### 4.2. Apigenin

Apigenin is a 4′, 5, 7-trihydroxyflavone with a chemical formula of C_15_H_10_O_5_ and a molecular weight of 270.24 g/mol. Apigenin is found in many plants, mainly grapes, apples, parsley, chamomile and red wine [96,97].

Treatment of prostate cancer cells PC-3 and DU145 with apigenin suppressed the apoptosis inhibitors XIAP, c-IAP1, c-IAP2 and survivin protein levels, and apoptosis occurred through a decrease in Bcl-xL and Bcl-2 and an increase in BAX along with an increase in cytochrome c (5, 10, 20, 40 μM) [32]. In addition, an increase in p21, a cancer suppressor protein, was shown in LNCaP cells (1, 5, 10 and 20 μM) [98]. In PC-3 and 22Rv1 cells, it inhibited IκB Kinase α (IKK α), which activates NF-κB, which regulates inflammation and cancer progression, and suppressed the activation of p65 (2.5, 5, 10 and 20 μM) [99]. Apigenin has also been shown to induce extrinsic apoptosis by upregulating the expression of caspase-8, -3 and TNF-a in PC-3 cells and prostate cancer stem cell CSC (CD44 (+)) isolated from there (25 μM) [100].

Apigenin was administered to 8-week-old male TRAMP mice at 20 or 50 μg/mouse/day (wt/vol) by gavage for 20 weeks. As a result, the development of prostate cancer through the PI3K/AKT/FOXO pathway was inhibited, and there was no toxicity or reduction of body weight found [101]. The following doses are similar to the daily intake of flavonoids in humans reported in previous studies [102]. PC-3 and 22Rv1 cells were transplanted into mice by subcutaneous injection in the flanks, and administration of the same amount of apigenin by gavage inhibits tumor growth, reduces tumor growth through inhibition of IKK phosphorylation and induces apoptosis. In this mouse model, treatment with two different concentrations of apigenin does not appear to cause adverse effects [99] (Table 1).

### 4.3. Acacetin

Acacetin is a 5, 7-dihydroxy-4′-methoxyflavone, chemical formula of C_16_H_12_O_5_ and a molecular weight of 284.26 g/mol [29]. Acacetin is mainly found in propolis, safflower and the *Asteraceae* plant family [103].

Acacetin decreased phospho-AKT and phospho-GSK-3β and increased the cancer suppressor p53 in DU145 prostate cancer cells (12.5 and 5 μM). In addition, the activation of phospho-IκB and NFκB was inhibited, and apoptosis was caused by the decrease of XIAP and Bcl-2 [29].

Acacetin (20, 30 and 50 μM) also inhibits cancer cell proliferation and cell growth by reducing the phosphorylation of STAT3 in DU145 cells and then induces apoptosis by suppressing the expression of STAT3 target proteins, such as Bcl-2, Bcl-xL, Mcl-1, cyclin D1 and survivin. Acacetin also showed anticancer activity through direct binding by interacting with the SH-2 domain, a domain of many signaling proteins, including the Src (steroid receptor coactivator) tumor protein of STAT3 [30].

Acacetin (25, 50, 100 μM) showed cell growth inhibition in LNCaP and DU145 cells and showed G1 or G2-M phase cell cycle arrest due to increased Ciap/p21 and the decrease of CDK2, CDK4 and CDK6. Higher levels of G2-M phase arrest occurred due to more Cdc25C, Cdc2/p34 and cyclin B1 reduction in LNCaP than in DU145 cells. In addition, apoptosis occurred due to the cleavage of poly-(ADP-ribose) polymerase (PARP) [104]. 

Acacetin (50 mg/kg) was intraperitoneally injected 5 days/week for 30 days into a BALB/c mouse model xenografted with DU145 cells. Tumor volume and weight decreased, and tumor growth was inhibited [30]. Zhou et al., reported on the potential for toxicity through cytochrome P450 inhibition when acacetin is supplied intraperitoneally at a 50 mg/kg dose to rats. However, there has yet to be a report on the specific toxicity. Thus more research is required [105] (Table 2).

### 4.4. Tangeretin

Tangeretin is a 4′, 5, 6, 7, 8-penta methoxyflavone and is mainly found in the citrus family. The chemical formula of this flavone is C_20_H_20_O_7_, and the molecular weight is 372.37 g/mol [106].

Tangeretin (20, 50, 75 and 100 μM) was treated with androgen-insensitive PC-3 cells, and androgen-sensitive LNCaP cells among prostate cancer cells, nuclear shrinkage, chromatin condensation and apoptotic bodies were formed. Especially in PC-3 cells, the pro-apoptotic markers BAX, caspase-3 and -9 were up-regulated, and the anti-apoptotic marker Bcl-2 was down-regulated. In particular, the phosphoinositide 3 kinase/protein kinase B/mammalian target of rapamycin (PI3K/Akt/mTOR) signaling pathway regulates this pathway as a critical regulator of apoptosis, cell growth, cell cycle and proliferation. It suggests a potential drug target for prostate cancer treatment [107]. In addition, epithelial-to-mesenchymal transition (EMT) plays a key role in tumors [108]. In PC-3 cells, tangeretin down-regulated mesenchymal proteins EMT markers, such as vimentin, CD-44 and N-cadherin, whereas up-regulated epithelial markers, such as E-cadherin and cytokeratin-19 [107]. In addition to PC-3 and LNCap, tangeretin decreased Bcl-2 and Bcl-xL and increased caspase-3, BAX and BAD in DU145, and apoptosis occurred through DNA fragmentation. Tangeretin was also found to inhibit androgen receptor (AR) and prostate-specific antigen (PSA) [109]. 

The results of single substance administration of tangeretin have yet to be published, and since tangeretin is also part of citrus extracts, similar in vivo effects to CPE are expected. Citrus peel extract (CPE), but not tangeretin, was injected intraperitoneally (25 μL and 50 μL) five times a week for 23 days into immunodeficient mice with human prostate cancer cell line PC-3 tumor xenografts. Tumors were mostly eliminated from mice injected with the high dose. In addition, both tumor weight and size were significantly reduced. Oral administration of 50 μL and 100 μL of CPE suppressed tumor growth, incidence and volume after 21 days [110]. Citrus peel extract (CPE) treatment by intraperitoneal injection and oral administration significantly decreased tumor weight and size without any observable toxicity, according to research examining its effectiveness against prostate cancer in a xenograft mouse model [111] (Table 3).
ijms-24-09240-t001_Table 1Table 1Anticancer effect of apigenin in vitro and in vivo.ApigeninProstate Cancer Cell LineTreatment ConcentrationAnticancer Regulation MechanismReferencePC-3 1, 2.5, 5, 10, 20, 25 μM■XIAP, c-IAP1, c-IAP2 ↓■Bcl-xL and Bcl-2 ↓■Cytochrome c and BAX ↑■IκB Kinase α ↓■p65 ↓■Caspase-3, -8 and TNF-a ↑[32]DU1451, 5, 10, 20 μM■XIAP, c-IAP1, c-IAP2 ↓■Bcl-xL and Bcl-2 ↓ ■Cytochrome c and BAX ↑[98]22Rv12.5, 5, 10, 20 μM■IκB Kinase α ↓■p65 ↓[99]LNCaP1, 5, 10, 20 μM■p21 ↑[98]CSC25 μM■Caspase-3, -8 and TNF-a ↑[100]**Mouse Model****Dosages****In Vivo Function**
8-week-old male TRAMP mice20 and 50 μg/mouse/day (wt/vol)/gavage for 20 weeks■PI3K/AKT/FOXO pathway ↓[101]Transplanted mice PC-3 and 22Rv1 cells 20 and 50 μg/mouse/day (wt/vol)/gavage for 20 weeks■IKK phosphorylation ↓■Apoptosis ↑[99]
ijms-24-09240-t002_Table 2Table 2Anticancer effect of acacetin in vitro and in vivo.AcacetinProstate Cancer Cell LineTreatment ConcentrationAnticancer Regulation MechanismReferenceDU1455, 12.5, 20,30, 50, 100 μM■Phospho-IκB and NFκB ↓■XIAP and Bcl-2 ↓■Phosphorylation of STAT3 ↓■Bcl-2, Bcl-xL, Mcl-1 and cyclin D1 ↓■Interacting with the SH-2 domain ↑[29,30]LNCaP25, 50, 100 μM■Ciap/p21 ↑■CDK2, CDK4 and CDK6 ↑■PARP ↑[104]**Mouse Model****Dosages****In Vivo Function**
DU145 xenografted BALB/c mouse model50 (mg/kg)/5 days/intraperitoneally injected/week for 30 days■Tumor volume, weight and growth ↓[30]
ijms-24-09240-t003_Table 3Table 3Anticancer effect of tangeretin in vitro and in vivo (CPE).TangeretinProstate Cancer Cell LineTreatment ConcentrationAnticancer Regulation MechanismReferencePC-320, 50, 75, 100 μM■BAX, caspase-3 and -9 ↑■Bcl-2, Bcl-xL ↓■PI3K/Akt/mTOR pathway ↓■Vimentin, CD-44 and N-cadherin ↓■E-cadherin and cytokeratin-19 ↑[107,108]LNCaP20, 50, 75, 100 μM■Nuclear shrinkage ↑■Chromatin condensation ↑■Apoptotic bodies ↑[107,108]DU14525, 50, 100 μM■Bcl-2 and Bcl-Xl ↓■Caspase-3, BAX and BAD ↑[109]**Mouse Model****Dosages****In Vivo Function**
PC3 xenografted immunodeficient mice25 μL and 50 μL/intraperitoneally injected/five times a week for 23 days■Tumor weight and size ↓[110]Oral administration of 50 μL and 100 μL■Tumor growth and volume ↓


## 5. Discussion

There has been a lot of research done to treat prostate cancer, and the treatment is likewise diverse.

Commonly, surgical removal of the prostate or hormone therapy to block or lower the amount of testosterone and are also adopts high-energy radiation therapy to kill prostate cancer cells are carried out. However, these therapies still have adverse effects and carry many risks [112,113,114]. Chemical agents, such as enzalutamide as well as apalutamide, the drug compared in this review, also have side effects that are not significantly different from the side effects of all chemical agents. In response to this, many studies have been conducted on the treatment of cancer using natural physiologically active substances of plants, such as flavonoids, and many studies that can actually be expected to be effective have been conducted [15]. Among them, flavones are effective against cancer, as discussed in this review, and their side effects are thought to be much less than those of other general anticancer drugs.

In this review, we selected apigenin, acacetin and tangeretin and confirmed that these three flavones are effective against prostate cancer through various apoptosis and anticancer pathways. In particular, since the regulation of apoptosis is an important mechanism to control cancer, this review focused on the regulation of prostate cancer through the apoptosis mechanism of three selected flavones [115].

Whether these three selected flavones are as effective as existing drugs, such as enzalutamide or apalutamide, can be found through various methods. Especially, molecular docking is widely used because it highly accurately determines structure-based drug de-sign, functional site prediction on the surface of protein molecules, and the shape of pro-tein-ligand binding sites [8].

For further study, we wish to select three flavones (apigenin, acacetin, tangeretin) and investigate their anti-cancer effects on prostate cancer. In addition, we will use molecular docking to predict the binding site and wish to present an excellent natural model of the anticancer effects of three flavones.

One thing to note is that the results shown in the molecular docking process are not analyzed by calculating the exact binding affinity. Here, the absolute docking result value is meaningless, and the primary purpose is to compare the result value between structures. In addition, even if a high result value is obtained, there are cases where the binding structure does not actually bind. Hence, the molecular docking result necessarily requires a process of visually confirming the structural bonding [116].

In other words, this study chose apalutamide as a comparison drug. However, the fact that the docking score alone does not necessarily mean that the three flavones are superior to or more efficient than this drug. In short, among many treatments for prostate cancer, it is possible to predict that flavones, which are natural physiologically active compounds in particular, have the conditions to be sufficiently effective as a treatment for prostate cancer.

## 6. Conclusions

In this review, we recalled that the three selected flavones, apigenin, acacetin and tangeretin, are effective in treating prostate cancer and show that they have sufficient value as a prostate cancer treating agent. Furthermore, more scientific evaluation on bio active compounds, such as flavonoids’ role in cancer, and their mechanisms is needed to develop new alternatives to treat the cancer in the future.

## Figures and Tables

**Figure 1 ijms-24-09240-f001:**
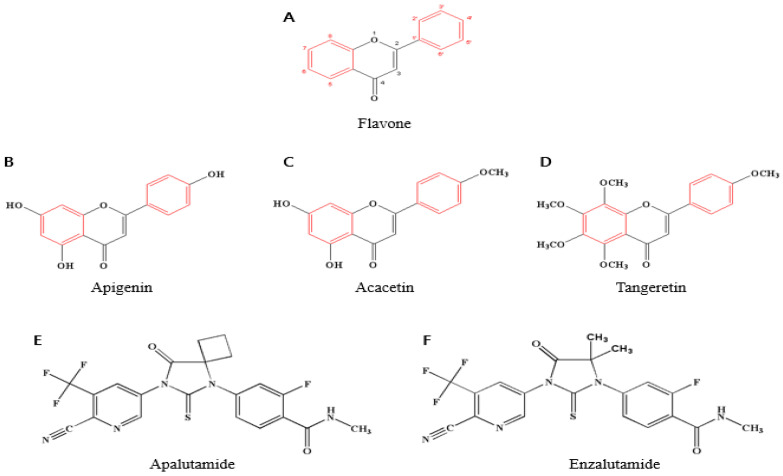
(**A**) The chemical structures of flavones (apigenin, acacetin, tangeretin), Apalutamide and Enzalutamide. The information on the chemical structure of (**B**) apigenin, (**C**) acacetin, (**D**) tangeretin, (**E**) Apalutamide and (**F**) Enzalutamide were from PubChem (https://pubchem.ncbi.nlm.nih.gov/, accessed on 3 February 2023). The red color is a typical structure derived from flavonoids and the numbers are carbon position. These images were created using ChemDraw Pro 8.0.

**Figure 2 ijms-24-09240-f002:**
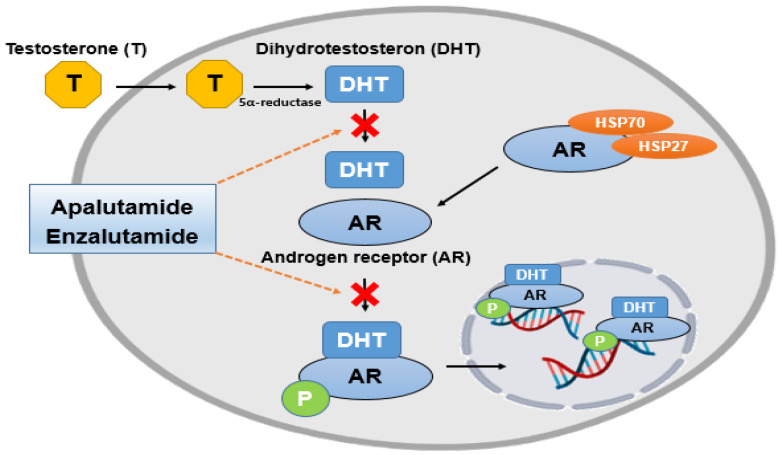
The mechanism by which two drugs, Enzalutamide and Apalutamide, prevent DHT from binding to AR, thereby counteracting its translocation to the nucleus. This image has been created using BioRender.

**Figure 3 ijms-24-09240-f003:**
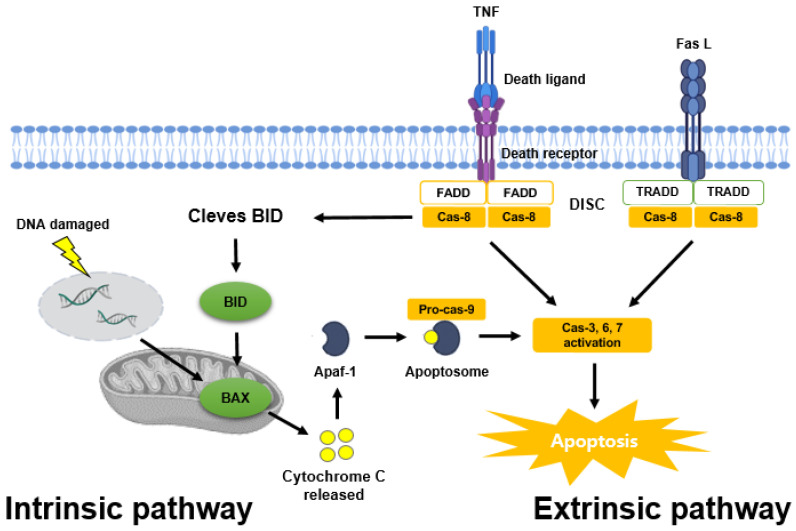
Two major apoptosis pathways (intrinsic and extrinsic cell death pathways). This image has been created using BioRender.

## Data Availability

Not applicable.

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
