# Peer review of "Flavones: The Apoptosis in Prostate Cancer of Three Flavones Selected as Therapeutic Candidate Models"

_ijms, 2023, doi:10.3390/ijms24119240_

Round 1

Reviewer 1 Report

In this review article, the authors examined the anticancer properties of flavones in prostate cancer. Focusing on three distinct flavones (Apigenin, Acacetin, Tangeretin), they reviewed the apoptotic-mediated effects in both in-vitro and in-vivo prostate cancer models. Additionally, they also performed comparative analysis of the molecular docking between androgen receptor, Apigenin, Acacetin, and Tangeretin. Interestingly, they reported that all three tested flavones are capable of occupying the active site of androgen receptor. Therefore, they proposed these molecules as possible candidates for the treatment of prostate cancer.  

On the whole, the article persuasively debated on the selected topic. Text, figures and table quality is acceptable. Concerning the references, although they are consistent with the proposed findings, outdated ones are used several times.

Major concerns:

Looking at IJMS guidelines, reviews are defined as article that should provide a complete and balanced overview of the latest progress in a given area of research. They should not merely summarize the literature or focus on authors' own work. Moreover, unpublished research data are not allowed. After reading the proposed article, one question occurred to me: are the authors proposing a review or an original article? Because the submitted one contains hybrid information. In the first part of the text, the authors provided an overview on both prostate cancer and flavones anticancer role in this specific tumor. However, in the second part they performed molecular docking analyses, proposing new data. Are these results already published? If so, this study can be considered as a review. Data must be referenced and material and methods section must be removed. If not, as already mentioned, reviews cannot contain original data. Therefore, the second part of the text should be removed.

Focusing on the review part, several aspects should be improved:

·         In the reported studies, these molecules were used at high concentrations in cell models (20-25 uM). It might be interesting to mention studies in which these molecules were tested in non-tumorigenic models with the purpose of speculating on their low toxicity against normal cells (if existing). As also mentioned in the article, current treatment has two main failures, namely acquired resistance and side effects. Proving a minimal body toxicity induced by flavones in animal models represent a relevant aspect that should to be mentioned, for instance.

·         Redundant aspects are often present across the text. Apoptosis pathways and Apoptosis in prostate cancer are quite similar, since the same molecular pathways are involved. Conversely, it can be usefel to discussion about mutations affecting apoptotic genes in prostate cancer.

·         Prostate cancer classification has only been reported in the discussion section. In my opinion, this aspect should be extensively debated in section 2 “Prostate Cancer”.

·         Most of this manuscript consists of simple restatements of other results’ papers. Very little is offered in the way of original analysis or opinion, which should be the entire point of writing a review paper.

Minor concerns:

·         Line 106: selenium has repeated two times. I suppose that one should be replaced.

·         The authors should declare how Figures (expressly figure 1,2 and 3) were taken or made (software, access time, …).

Dear Editor,

Thank you so much for inviting me to review the current manuscript. it's an honor to make my personal contribution in improving the relevance and the scientific impact of your valuable journal.   

In my view, this study is not ready for publication in its present form. I suggest rejecting and resubmitting once appropriate changes have been made. For more information about my decision, please see the Comments and Suggestions for Authors.

Once again thanks for considering me a suitable reviewer for your valuable journal. 

Author Response

Reviewer 1

  1. References outdated

Response- Thank you for your comment. While we referring for the review content, we looked for the original text of the associated content, rather than other review papers, and the old paper was referred to.

  1. Is it a review or a research paper?

Response- Thank you. It is a review article. Molecular docking itself is used as a tool to show the degree of binding between two molecules or between a receptor and a ligand-based on existing data. Therefore, molecular docking using existing data is only one way to present my original analysis and opinion on the effect of three flavones on prostate cancer. Many studies have shown that all three flavones are already effective against several cancers or prostate cancer. I would appreciate it if you would consider that we presented original analysis using a kind of big data called molecular docking that there is this same effect in terms of structural bonding.

Here, include previous papers that used similar kind of combination of review along with original data. (Essa, A.F., Teleb, M., El-Kersh, D.M. et al. Natural acylated flavonoids: their chemistry and biological merits in context to molecular docking studies. Phytochem Rev (2022). https://doi.org/10.1007/s11101-022-09840-1)

In order to fit the review format as much as possible in relation to molecular docking, the material and method section was deleted, and the program used, the ID number of molecular data, access time, etc., were written under figures 4, 5, 6, and 7.

I would appreciate it if you could think again about these formatting and opinion changes.

  1. Body toxicity in mouse model

Response- Thank you for a suggestion. Toxicity information for each flavone in the mouse model was added at the end of each of 3.1.1, 3.1.2 and 3.1.3.

“The following doses are similar to the daily intake of flavonoids in humans reported in previous studies.” Line 287-288

“In this mouse model, treatment with two different concentrations of apigenin does not appear to cause adverse effects.” Line 291-293

“Zhou et al. reported on the potential for toxicity through cytochrome P450 inhibition when acacetin is supplied intraperitoneally at a 50 mg/kg dose to rats. However, there has yet to be a report on the specific toxicity. Thus more research is required.” Line 315-318

Article citation like mention below.

(“Anwanwan, D., Singh, S. K., Singh, S., Saikam, V., & Singh, R. (2020). Challenges in liver cancer and possible treatment approaches. Biochimica et Biophysica Acta (BBA)-Reviews on Cancer, 1873(1), 188314.”)

“Citrus peel extract (CPE) treatment by intraperitoneal injection and oral administration significantly decreased tumor weight and size without any observable toxicity, according to research examining its effectiveness against prostate cancer in a xenograft mouse model.” Line 345-348

  1. Mutations affecting apoptosis genes in prostate cancer

Response- Thank you. We have included “The genetic mutations of apoptosis in prostate cancer” section to 2.5.3 of this manuscript. Line 252-262

  1. The classification of prostate cancer is extensively discussed in section 2

Response- Thank you. We have included a paragraph on the classification of prostate cancer in 2.2 of the main text for extensive discussion. Line 141-160

  1. Original analysis or opinion

Response- Thank you. Our primary focus was to determine the comparison using molecular docking after selecting three flavonoids from numerous flavonoids through an extensive literature search can be unique approach of this manuscript.

 In order to select three flavones effective for prostate cancer, a lot of literature search was done, and the effect on prostate cancer was investigated through the selected three, and it is true that there are restatements of other results. However, I think it is original to show results from a new aspect through the content of molecular docking. I would be grateful if you could think about this for a moment.

  1. Selenium has been repeated two times

Response- Thank you for your correction. We have corrected it in a revised manuscript.

 “A deficiency of vitamin D increases the risk of prostate cancer, and supplements such as selenium reduce the risk of prostate cancer.” Line 122-123.

  1. How Figures 1, 2, and 3 were created (software, access time, etc.)

Response- Thank you. We have included software and access time used under each figure.

Lastly, English proofreading was completed with a native speaker.

Again, we thank you for your thoughtful comments.

Reviewer 2 Report

I would like to make some suggestions for the improvement of this manuscript.

1. In the formula of the general structure of flavone, the numbering at the oxygen heteroatom, position 1, is missing.

2. the structural formula should also be specified of enzalutamide line 148

3. In subsection 2.1. lines 98-108 it would be useful to compare the benefits and risks to dissociate them, since at the moment they are mixed. In addition, selenium is written twice in line 106.

4. It would be useful to tell some side effects of the 2 drugs, compared to the benefits of the 3 flavones. From this point of view, the importance of the 3 flavones would stand out.

5. Most helpful for the general public to understand the contents of this manuscript, I suggest defining the abbreviations by making a list of them, as they are spread throughout the material.

6. A last suggestion to delimit the discussions from the conclusions

7. or each reference should be put [Google Scholar] [Cross Ref] for example.

Author Response

Reviewer 2

  1. In the formula of the general structure of flavone, the numbering at the oxygen heteroatom, position 1, is missing.

Response- Thank you for your correction. We have corrected basic flavone structural formula of Figure 1, the oxygen heteroatom number 1 in the revised manuscript. (Figure 1).

  1. The structural formula should also be specified of enzalutamide line 148

Response- Thank you for your comment. We have added the structural formula of enzalutamide to Figure 1. Enzalutamide (Figure 1. F). Line 186.

  1. In subsection 2.1. lines 98-108 it would be useful to compare the benefits and risks to dissociate them, since at the moment they are mixed. In addition, selenium is written twice in line 106.

Response- Thank you for your comment. In subsection 2.1, the dietary benefits and risks of prostate cancer were separated and modified in the revised manuscript.

  1. It would be useful to tell some side effects of the 2 drugs, compared to the benefits of the 3 flavones. From this point of view, the importance of the 3 flavones would stand out.

Response- Thank you. In subsection 2.4, we have added the side effects of the two drugs, and in the discussion subsection 5.1, we updated brief discussion of the flavonoids benefits compared to the two chemical drugs. Line 437-444

  1. Most helpful for the general public to understand the contents of this manuscript, I suggest defining the abbreviations by making a list of them, as they are spread throughout the material.

Response- Thank you for your suggestion. A list of abbreviations was created and added to the first page. Line 32-46

  1. A last suggestion to delimit the discussions from the conclusions

Response- Thank you. The discussion and conclusion were separated and revised in the updated manuscript.

  1. Each reference should be put [Google Scholar] [Cross Ref] for example.

Response- Thank you. All the references are collected from Google Scholar in APA format and cited by using the Endnote tool. The link for Google Scholar or Cross Ref will be attached by the publisher once the paper got accepted.

Lastly, English proofreading was completed with a native speaker.

Again, we thank you for your thoughtful comments.

Round 2

Reviewer 1 Report

After the first round revision, the authors submitted a revised version of the manuscript and the related point-by-point response letter.

In my opinion, the original article is improved since the authors have embraced most of the proposed concerns.

Nevertheless, the authors decided to maintain the molecular docking analysis, that is basically considered as original data. Looking at IJMS guidelines, reviews are defined as article that should provide a complete and balanced overview of the latest progress in a given area of research. They should not merely summarize the literature or focus on authors' own work. Moreover, unpublished research data are not allowed.